# MAGNITUDE: Transcranial Magnetic Stimulation for Treatment-Resistant Obsessive–Compulsive Disorder: A Randomized Sham-Controlled Phase II Trial Protocol

**DOI:** 10.3390/brainsci15020106

**Published:** 2025-01-23

**Authors:** Lavinia Rech, Ricardo A. Vivanco, Ana Claudia Guersoni, Gianina M. Crisóstmono Ninapaytan, Paulina Bonilla Rivera, Elisabeth J. Ramos-Orosco, Ariana Vargas-Ruiz, Martha Felipe, Sandra Carvalho

**Affiliations:** 1ECPE Department-PPCR Program, Harvard T. H. Chan School of Public Health, Boston, MA 02115, USA; 2Division of Cardiology, University Heart Center Graz, Medical University of Graz, 8036 Graz, Austria; 3Department of Neurology, University of Kentucky, Lexington, KY 40536, USA; 4Faculty of Medicine, Preventive Medicine Department, University of São Paulo, São Paulo 01246 903, Brazil; 5Instituto Nacional de Salud Mental “Honorio Delgado—Hideyo Noguchi”, Universidad Peruana Cayetano Heredia, Lima 15102, Peru; 6Hospital del Trabajador, Asociación Chilena de Seguridad ACHS, Santiago 7501239, Chile; 7Department of Pediatrics, Nationwide Children’s Hospital, The Ohio State University, Columbus, OH 43205, USA; 8Children’s National Hospital, San José 267-1005, Costa Rica; 9School of Global Public Health, New York University, New York, NY 10012, USA; 10Psychological Neuroscience Laboratory, Basic Psychology Department, Centro de Investigação em Psicologia (CIPsi), School of Psychology, University of Minho, 4710-057 Braga, Portugal

**Keywords:** OCD, obsessive–compulsive disorder, treatment-resistant, rTMS, transcranial magnetic stimulation, dLPFC, dorsolateral prefrontal cortex

## Abstract

Obsessive–Compulsive Disorder (OCD) is a chronic psychiatric condition with a lifetime prevalence of 2–3%. It significantly burdens quality of life and is associated with substantial economic and disease burdens. Cognitive-behavioral therapy and high-dose selective serotonin reuptake inhibitors are considered the first-line treatments for OCD. Approximately two-thirds of patients with Obsessive–Compulsive Disorder (OCD) exhibit inadequate responses to current standard therapies, thus lacking adequate therapy, resulting in a loss of quality of life and huge economic burdens. Repetitive transcranial stimulation (rTMS) is a non-invasive, safe, and well-tolerated intervention that modulates prefrontal cortical circuits involved in OCD. A previous systematic review explored the therapeutic effects of rTMS applied to the dorsolateral prefrontal cortex (dlPFC) area in patients with treatment-resistant OCD. It showed that the application of high-frequency and low-frequency (LF) rTMS to the dlPFC region yielded controversial post-treatment Y-BOCS (Yale-Brown Obsessive–Compulsive Scale) findings due to factors such as small sample sizes, short-term study durations, and variations in rTMS protocols. Objectives: Thus, we propose a theoretical protocol based on previous findings to assess better the effect of LF rTMS for treatment-resistant OCD patients. Methods: We will recruit patients with moderate to severe OCD and limited response to previous treatments from in- and outpatient clinics. We will use fMRI for precious localization of the right dlPFC and application of 1 Hz stimulation of in total 2000 pulses with three times 40 s inter-train intervals 5 days a week, in 6 consecutive weeks. The primary outcome will be the mean reduction in Y-BOCS at the end of this study. Conclusions: This study highlights rTMS’s potential to reform OCD treatment, accentuate safety, accessibility, clinical integration, and future research foundations.

## 1. Introduction

Obsessive–Compulsive Disorder (OCD) is a chronic psychiatric condition characterized by intrusive, repetitive, and distressing thoughts (obsessions), as well as repetitive behaviors or mental acts (compulsions)—which may be overt or covert—that often consume a significant amount of time. These compulsions are typically carried out to neutralize the fear and anxiety triggered by the obsessions, and the disorder is frequently accompanied by varying levels of insight [1].

With a lifetime prevalence of 2–3%, OCD imposes a significant burden on quality of life and is associated with substantial economic and disease burdens [2,3,4]. First-line standard therapy for OCD patients is cognitive-behavioral therapy (CBT) with or without additional selective serotonin reuptake inhibitors (SSRIs). However, 20–60% of these patients experience non-response to these first-line therapies [5]. Following this, second-line treatment would include the interchange of SSRI, Clomipramine, or serotonin noradrenalin reuptake inhibitors (SNRI), with additional add-on treatments [6]. Still less than 10% are currently receiving evidenced-based treatments, highlighting the need for alternative treatments and moving forward to interventional options [6]. The situation of untreated or neglected patients underscores the urgency to explore safe therapeutic alternatives [7].

From a mechanistic standpoint, OCD is closely linked to dysregulation within cortico-striato-thalamo-cortical (CSTC) circuits, which encompass the orbitofrontal cortex, striatum, thalamus, and related subcortical structures [8]. Within this circuitry, the dorsolateral prefrontal cortex (dlPFC) plays a pivotal role in regulating working memory and executive function. Aberrations in front-limbic networks, including those involving the dlPFC, contribute to maladaptive fear responses, intolerance of uncertainty, and the persistence of compulsive behaviors [9,10,11]. Despite ongoing refinements, prevailing neurocircuit models often fail to fully account for the complexity of OCD, including co-occurring psychiatric conditions and the dynamic, longitudinal course of symptoms. In this context, Gonaçalves et al. (2011) offered a medical hypothesis proposing that OCD may arise in part from interhemispheric imbalances in CSTC circuits, potentially linked to difficulties in sensory and cognitive filtering at the thalamic level. According to this perspective, effective interventions—particularly those involving cognitive-behavioral therapy or neuromodulatory approaches—could not only alleviate OCD symptoms but also induce measurable neuroplastic changes in these circuits. Non-invasive brain stimulation (NIBS) approaches possess the capacity to restore functional interhemispheric balance by modulating the excitability and connectivity of specific cortical regions. This comprehensive perspective highlights the necessity for a more sophisticated and dynamic comprehension of OCD pathophysiology, transcending static models and integrating the brain’s intrinsic ability for adaptive remodeling in response to treatment [11].

However, existing neurocircuit models often neglect co-occurring psychiatric conditions and longitudinal changes in symptoms, warranting a comprehensive approach to understanding and treating OCD. Repetitive transcranial magnetic stimulation (rTMS) is a non-invasive neurostimulation technique that modulates neuronal activity via magnetic pulses delivered by a wire coil through the scalp, leading to long-lasting effects on the brain [12]. Due to its impact on chronic brain disorders, several TMS devices have been FDA-approved for treatment-resistant major depressive disorder since 2008, and new indications are currently under development [13].

rTMS exerts selective influence over specific brain regions, using different stimulation protocols with various physiologic purposes [12]. Generally speaking, low-frequency (LF) rTMS inhibits, and high-frequency (HF) rTMS excites. Assessing neurostimulation response is challenging due to potential effects on surrounding areas and circuits, complicating clinical evaluation [9]. Studies targeting dlPFC in major depression disorder (MDD) reveal varied excitability in regions like the somatosensory cortex, subgenual anterior cingulate cortex, superior parietal lobe, and temporal regions [14].

Current research on rTMS in OCD lacks a definitive protocol for treatment-resistant patients. A meta-analysis by Fitzsimmons et al. displayed the medial frontal cortex and the dlPFC as promising options [15]. A more specific network analysis of the different protocols revealed significant effectiveness for LF right dlPCF stimulation, HF bilateral dlPFC stimulation, and LF stimulation of the pre-supplementary motor area (pre-SMA). Additionally, the accelerated continuous theta-burst stimulation (cTBS), a specific kind of rapidly fast rTMS, on the SMA in treatment-resistant OCD patients significantly improves psychopathology, severity of illness, and depression among those patients [16]. All studies showed the common side effects of rTMS, including headaches, scalp discomfort, muscle spasms, mood changes, and less common seizures [17]. These are mostly mild and reversible, suggesting that rTMS is safe. Moreover, the FDA approved a TMS system by Brainsway for OCD based on the lack of serious adverse events in one revised trial [2,18].

However, this approved alternative is for deep TMS and, therefore, requires a specific coil to stimulate deep brain areas, which limits its availability in a wide real-world setting [19]. Our theoretical trial will focus on rTMS, as the needed coils will be more available. While different targets are potentially effectual, the right dlPFC seems the most effective in the general OCD population [15,20]. Unfortunately, conflicting evidence exists in treatment-resistant patients [21]. Moreover, the standard assessment of the optimal localization of stimulation of the dlPFC lacks accuracy. A study by Zhang et al. found that the dlPFC, assessed by the 5 cm rule, was meticulous in 13.8% to 54% [22]. Therefore, there is a need for an MRI-based neuronavigation to localize the dlPFC exactly individually to reduce side effects in other areas and improve the efficient investigation of treatment in this area.

Consequently, this theoretical trial seeks to determine the efficacy and safety of low-frequency repetitive transcranial magnetic stimulation (LF rTMS) applied to the right dorsolateral prefrontal cortex (dlPFC) in conjunction with standard therapy for treatment-resistant Obsessive–Compulsive Disorder (OCD) in adults. The hypothesis postulates that the rTMS protocol will result in a statistically significant reduction in Yale-Brown Obsessive Compulsive Scale (Y-BOCS) scores compared to a sham intervention in patients undergoing standard therapy over 6 weeks.

## 2. Methods

### 2.1. Trial Design

This trial will be a phase II multicentric, sham-controlled, parallel trial. Eligible patients will be randomly assigned in a 1:1 ratio to receive either low-frequency rTMS of the right dlPFC or sham rTMS, both in conjunction with standard therapy.

Given the relatively low prevalence of treatment-resistant OCD, patient recruitment will be conducted across multiple outpatient psychiatric centers in the United States to ensure adequate participant enrollment.

### 2.2. Randomization

#### 2.2.1. Randomization Sequence Generation

Eligible participants who provide informed consent after the baseline visit will be randomized and assigned in a 1:1 ratio to one of the two intervention groups, stratified by site. Randomization will be conducted using a computerized, web-based system called Interactive Response Technology (IRT). This approach ensures a balanced distribution of key covariates across the study sites. The centralized IRT system will generate the allocation sequence and assign the code corresponding to either active or sham rTMS. The study coordinator will implement the allocation specified by the IRT system.

Each site’s delegated physician or clinical research investigator will discuss the trial information with participants and obtain and document informed consent. Prior to any study visit or procedure, participants willing to participate will receive comprehensive information about the study and complete the informed consent process. During the screening visit, participants will have at least 24 h to review the study information before providing consent. After verifying eligibility criteria and obtaining informed consent, investigators will enroll the participants in the study and proceed with randomization.

#### 2.2.2. Allocation Concealment

At each study site, a designed person will conduct pre-screening and assess participant eligibility following the competition of the informed consent process. This person is responsible for initiating the randomization procedure. Therapy assignments will remain concealed until the investigator performing the randomization has: 1. identified themselves (i.e., verified their identity within the randomization system), 2. confirmed the participant’s eligibility (i.e., ensured that the participant definitely meets all inclusion criteria), and 3. documented confirmation (i.e., recorded the eligibility confirmation in written form to the central randomization office through the REDCap platform (https://www.project-redcap.org, accessed on 25 November 2024). This approach ensures that the allocation concealment is maintained, thereby preventing selection bias and ensuring the integrity of the randomization process.

#### 2.2.3. Implementation

The implementation of randomization is strictly limited to trained and designated professionals who uphold the integrity of the study, irrespective of researchers’ preferences or participants’ conditions. An independent researcher will generate the allocation sequence by creating randomization cards using computer-generated random numbers. The original allocation sequences will be securely stored in an inaccessible location, and only copies of these sequences will be used during the study to prevent unauthorized access or alterations. To avoid confusion between treatment codes (e.g., A and B), the allocator will maintain a detailed and secure record that maps each code to its corresponding treatment (rTMS or sham). This ensures that the executing technicians can accurately implement the treatment assignments without ambiguity. By centralizing the randomization process and restricting access to the allocation sequence, the study maintains robust control over the assignment process, thereby enhancing the reliability and validity of the study outcomes.

### 2.3. Blinding

#### 2.3.1. Blinding

To minimize bias and ensure the study’s integrity, participants and researchers will be blinded to the treatment assignments. Blinding will be achieved through the use of a purpose-built sham TMS coil equipped with a magnetic shield that attenuates the TMS effects. Additionally, surface electrodes will be added to stimulate the skin during a TMS pulse, producing somatosensory effects in the skin and peripheral nerves similar to those from an intervention. This combination ensures that participants cannot discern whether they are receiving active or sham treatment based on sensory feedback. All individuals involved in the study—including participants, healthcare providers, assessment researchers, and data analysts—will remain blinded to the treatment allocations throughout the study to reduce potential biases. Each participant will receive a unique code, to facilitate data analysis without revealing information about the treatment groups. This method helps to maintain objectivity in the evaluation outcomes. The executing technician tasked with setting up the treatment will be aware of the specifics but will not engage with other participants, including the healthcare provider or the study participants.

A third person who will serve as clinical sub-investigator will be responsible for participants’ enrollment and conduction of follow-up visits complying with the criteria of Good Clinical Practices and Human Subjects Research. Each center will require some clinical sub-investigators who will receive relevant protocol information and complete training. In case of minor adverse events, these can be discussed with the principal investigators without the need for unblinding. Participants will be unblinded 6 months after the end of the study. This delayed unblinding ensures that the primary data collection and analysis phases are free from bias while still providing participants with information about their treatment allocation once the study objectives have been met. By implementing these rigorous blinding procedures, the study aims to ensure an objective assessment of treatment effects, enhance the reliability of the findings, and uphold the highest standards.

#### 2.3.2. Emergency Unblinding

Patients must be unblinded to provide the best possible care if serious neurological or psychiatric side effects or adverse events such as sudden and significant changes in mood, thinking, or neurological function occur after the rTMS procedure. These include seizures or neurological deficits, an increase in severity of depression (marked by changes in the HDRS ≥ 25), and manic symptoms like psychotic symptoms. Further changing warning signs of suicidal ideation, based on the nomenclature from the National Institute of Mental Health, like talking about the want to die or feeling unbearable pain [23], will lead us to assess the suicidal ideation attribution with the SIDAS score [24]. In case of a score ≥21, this will also lead to unblinding.

#### 2.3.3. Blinding Assessment

The providers, participants, research assessors, and data analysts’ blinding will be assessed after 3 weeks and, ultimately, after 6 weeks of treatment analyzed with Bang’s blinding index [25].

### 2.4. Participants Eligibility Criteria

#### 2.4.1. Inclusion Criteria

Adults between 18 and 65 years old;Current obsessive–compulsive disorder, as diagnosed by current DSM5 criteria by a licensed psychiatrist;Moderate to Severe OCD score on the Y-BOCS of ≥16 as a clinically significant cut-off [26];Stable maintenance treatment for at least 12 weeks;Limited response (reduction inferior to 25% at YBOCS) to previous treatment defined as patients on maintenance treatment either with cognitive-behavioral therapy and/or SSRI and previous failure to respond to at least one SSRI [6].

#### 2.4.2. Exclusion Criteria

History of stroke, seizures, epilepsy, or head injury;The presence of implanted ferromagnetic or magnetic-sensitive devices in the head or neck;Pregnant or lactating women;Previous treatment with electroconvulsive therapy;Current or previous treatment with rTMS;Presence, history or diagnosis of psychosis, substance abuse, suicide attempt, bipolar disorder, ADHD, or schizophrenia, severe major depression disorder (HDRS ≥ 25);Patients with any type of cardiac arrhythmia and cardiac devices.

### 2.5. Recruitment Strategy

Subjects are enrolled in the study at each center based on the abovementioned inclusion and exclusion criteria. Patients seen in the inpatient and outpatient settings will be consecutively recruited in the study. Furthermore, we will contact psychiatrists and psychologists through regular mail and use advertisements on social media to recruit additional patients. The recruitment process will continue until the target population (55 participants per group; see “Sample size” section) is reached. The recruitment period will extend over 6 months. A team of research assistants will be responsible for responding to any inquiries about participation via phone and email. The team of researchers will conduct the recruitment interviews through videoconferencing or in-person meetings on their sites, ensuring the IRB guidelines will be followed.

### 2.6. Adherence

To maintain adherence during the intervention, we will send weekly weekend reminders via messaging apps for the week ahead. We will ensure appointments are at similar, convenient times for each subject and at an accessible local site. If necessary, we will offer to liaise with the subject’s carers. All participants will receive a 200 USD incentive, and vouchers for transport (e.g., parking) and food will also be provided. A log of subject attendance will be used to monitor adherence [27].

### 2.7. Interventions

#### 2.7.1. Interventions

Eligible patients will be randomized equally to either low-frequency transcranial magnetic stimulation of the right dlPFC or sham. TMS is a neuromodulation technique in which a stimulator sends electromagnetic pulses to the patient’s scalp through a coil. We will use a MagVenture MagPro X100 stimulator with two dynamically cooled butterfly B70 coils: real and sham (MagVenture, Farum, Denmark).

This trial will execute an inhibitory modulation of the target area using a low-frequency protocol, defined as a pulse rate of 1 Hz. The intensity refers to the power of the pulse and is determined based on the motor threshold (MT) explained later. Lastly, the number of pulses per session will determine the duration of the session [28].

Measuring MT for each patient is required to set the adequate stimulation intensity before the TMS sessions. It will be measured for each patient once, regardless of the randomized group. The method for calculating the MT will consist of using the thumb-movement visualization. First, the primary motor cortex of the hand (M1 hand) is localized by using the C1/C2 position of the 10–20 EEG system [29]. Single pulses are applied to the M1 hand with an interpulse interval (IPI) of at least 7 s to prevent additive neuromodulatory effects. The stimulation intensity is sequentially reduced until we reach a point when fewer than 50% of the pulses lead to a muscle contraction of the contralateral hand (identified by visual inspection or neurophysiological motor-evoked potentials). The first TMS intensity that cannot elicit a muscle contraction of more than 3 out of 6 pulses is considered the motor threshold, which is the percentage of the maximum stimulator output.

Identifying the localization of stimulation to inhibit the dlPFC in each patient is critical for the efficacy of TMS. Previous trials have used the 5 cm rule to localize the stimulation site by evoking contralateral thumb movement using TMS in the motor cortex. In this way, the stimulation site is determined as 5 cm anterior to and in a parasagittal line from the point of the maximum stimulation of the motor cortex [30]. Nonetheless, this approach has been inaccurate, considering systematic variability in anatomic landmarks between patients [31]. Neuronavigation technology offers more precise and accurate TMS coil positioning than the rule of 5 method [32]. Using real-time MRI imaging, we will use the Brainsight Neuronavigation system (https://brainbox-neuro.com/products/brainsight-tms-navigation, accessed on 16 December 2023) to localize the adequate coil position.

We then will use an inhibitory protocol of 1 Hz stimulation at 100% of the MT consisting of four trains of 500 pulses, each with a 40 s inter-train interval, allowing the coil to cool. It will be a total of 2000 pulses per session with a duration of 36 min, based on the parameters by Elbeh et al. [33]. This protocol complies with published safety guidelines [28]. Sessions will occur daily, Monday through Friday, over six consecutive weeks. Sham TMS will consist of the same procedure but using the sham coil. The sham coil will be positioned in the stimulation site of dlPFC using the neuronavigation system. It will also produce similar non-specific auditory and somatosensory effects but will be equipped with a magnetic shield that attenuates the magnetic field.

#### 2.7.2. Modification/Discontinuation

Transcranial magnetic stimulation is generally safe and produces minor adverse effects or, rarely, more severe adverse effects such as seizures. The intervention protocol will follow the recommendations of the Updated Expert Safety Guidelines for TMS [34]. Adequate measures will be taken to guarantee the safety of every patient in the trial.

##### Seizures

Inducing seizures is considered a severe adverse event, but the risk associated with conventional rTMS is low, with an estimated standardized risk of 8/100,000 sessions [28]. The proposed parameters for the rTMS protocol are below the minimal parameters for seizure induction. Additionally, we will exclude patients with an increased threshold of seizures as specified in the exclusion criteria. In the event of a seizure, the rTMS will stop immediately, and the patient will be stabilized with appropriate anticonvulsants and then transferred to the nearest specialized unit for further evaluation. After a thorough investigation, the event will be reported to the Principal Investigator, the Study Safety Board, and the Institutional Review Board.

##### Hearing Loss

There is a risk of transient increases in auditory thresholds, decreasing hearing sensation. Patients with previous auditory conditions, such as individuals with cochlear implants, will be excluded from the trial. Everyone should use appropriate hearing protection, such as earplugs, during the intervention session. Any patient complaining of hearing loss, tinnitus, or aural fullness will be referred for auditory evaluation.

##### Other Minor Adverse Events

rTMS may produce minor side effects such as muscle twitching and headaches. Patients will be warned about these transient and mild symptoms before each session. Headaches usually happen after the intervention and are associated with prolonged and fixed head positioning. Patients will receive over-the-counter analgesics in the case of headaches.

### 2.8. Outcomes

#### 2.8.1. Primary Outcome Measures

Our study’s primary outcome will be comparing the groups’ Y-BOCS scores. The Y-BOCS is a validated 10-item scale, rated by trained clinicians, that evaluates the severity of symptoms in patients with OCD. Each item is rated from 0 (less severe) to 4 (more severe), with a total score of 40 [35]. Patients will be evaluated with the Y-BOCS at baseline, two, four, and six weeks of the intervention.

#### 2.8.2. Secondary Outcome Measures

Secondary outcomes include the difference in response rate and the time-to-response between the groups. The response rate will be measured as a dichotomous variable to quantify the patients who achieve clinically significant improvements. A complete and partial response rate will be defined as a reduction of ≥35% in the Y-BOCS score [36]. The latter will be defined as the time at which a partial response of a reduction of ≥25% in the Y-BOCS score will be seen.

In addition, the Clinical Global Impression (CGI) scale will be used by an experienced psychiatrist to measure the global judgment of improvement or worsening. During the assessment of the CGI scales, the psychiatrist will judge the illness severity, the patient’s level of distress, and the impact on daily functioning [37]. Moreover, the Sheehan Disability Scale (SDS) will measure the degree of disability of the patient. The SDS is a three-item self-rate scale that provides insights into the level of impairment in work, social, and family functioning [38]. We will also measure self-reported improvement in symptoms of obsessions and compulsions utilizing the Padua Inventory (PI) [39]. Lastly, patients with OCD often have depression, which could be a significant moderator of the efficacy of TMS in these patients [40]. We will measure the depressive symptoms using the widely validated Hamilton Depression Rating Scale (HDRS). The HDRS 17-item clinician-administered tool is used to measure depressive symptoms [41]. All scales will be evaluated at baseline in two, four, and six weeks.

OCD patients display abnormal regional homogeneity (ReHo) in functional MRI (fMRI) in the right dlPFC region [42]. Therefore, an additional fMRI will be conducted during the baseline MRI to localize the right dlPFC. After 6 weeks of treatment, we will assess the post-treatment fMRI.

### 2.9. Data Management

#### 2.9.1. Data Monitoring

An independent Data Monitoring Committee (DMC) of external experts in the area, without conflict of interest relevant to this study, will be established. The DMC will assess the progress, safety data and, if needed, critical efficacy endpoints. Confidentiality will be maintained during data monitoring, review, and deliberations. The Project Officer (PO) and the DMC will develop meeting agendas. Voting and minutes will be kept confidential. Open Session and Close Section reports will be prepared, presented, and appropriately distributed. Recommendations on the trial’s continuity, pause, or termination and necessary adjustments will be tracked to ensure prompt and effective implementation.

All data will be entered electronically in the trial. The original study forms will be entered and kept on file at the participating sites. The Clinical Data Management System (CDMS) will handle the data. At every visit, an electronic version of the Case Report Form (e-CRF) will be applied to all participants. The collected data from the participating sites will be sent to the Core Coordinating Center.

The entries of the e-CRFs will be monitored by the Clinical Research Associate (CRA) for completeness, and filled-up CRFs will be retrieved and handed over to the Clinical Data Management (CDM) team. The CDM team will track the retrieved CRFs and maintain their record. An Edit Check Program will be developed to detect discrepancies after validation with dummy data. In the case of data discrepancy, a Data Clarification Form (DCF) will be generated and delivered to the responsible investigator, who will have 48 h to resolve the issue. To enforce the data integrity, the CDM team will review all discrepancies regularly to ensure they have been resolved. There will be a discrepancy database where all discrepancies will be recorded and stored with an audit trail.

The study data will have restricted access. Each participating site will only be able to access its own data. The access password will be renewed regularly. All reports prepared by the Data Coordinating Center (DCC) will be prepared so that no individual subject can be identified. A complete backup of the primary DCC database will be performed once every two weeks. Incremental data back-ups will be performed daily. Errors identified by the system will be summarized and communicated to Data Managers for verification and correction. De-identified data will be made available upon reasonable request, subject to all applicable privacy laws and regulations. These tapes will be kept off-site until the study is completed. Participant files are to be stored in numerical order and will be maintained in storage for 3 years after completion of the study. Upon completion of the study, the data will be archived on secure servers for at least 15 years, as required by regulatory guidelines. De-identified data can be shared upon request, ensuring compliance with relevant data privacy regulations.

#### 2.9.2. Interim Analysis

During the interim analysis for adverse effects at 3 weeks, a Data and Safety Monitoring Board (DSMB) will ensure patient safety and determine the continuation of the trial. The DSMB will compare the rate of adverse effects, especially serious events like seizures, between the groups. Because there will be no interim statistical analysis, there is no need for alpha spending function adjustments.

### 2.10. Sample Size Calculation

Due to the study design, the primary outcome will be evaluated by a two-sided *t*-test after 6 weeks of treatment. Values from Jahanbakhsh et al. [43] were used for sample calculations, namely Y-BOCS values of 27.40 ± 4.91 (control) and 27.53 ± 4.61 (treatment) at baseline and 27.33 ± 4.15 (control) and 24.07 ± 4.35 (treatment) post-treatment. Performing the sample size calculation (STATA BE 17.0) with an alpha of 0.05 and a power of 0.9, to reduce the type II error, estimated a size of 76 participants. Due to the low adverse events in rTMS, it has a low dropout rate, from 5.3% in active use to 11.28% in sham use [44]. Additionally, the OCD patient cohort has a various dropout rate, from 17.29% in pharmacological trials, over 20.63% in active control like metacognitive therapy, to 23.49% in pill placebo [45]. Considering these two, we aim for a 30% dropout rate to ensure we do not lose power during the trial. Thus, our calculation leads to 110 participants overall, with a sample size of 55 participants per group.

### 2.11. Statistical Analysis for Primary and Secondary Outcomes

Baseline characteristics will be presented as frequencies with percentages for categorical data, or means ± SD for data with normal distribution, respectively, and medians with interquartile range (IQR) for data with non-normal distribution. Statistical analyses will be evaluated as intention-to-treat. The primary endpoint of Y-BOCS as an ordinal variable on a 40-point Likert scale will be analyzed using a *t*-test [46] and Mann–Whitney-U test, respectively, and will be adjusted to sex, depression, and baseline severity of OCD with multiple regression. Normality will be assessed using the QQ-Plot and Kolmogorov–Smirnov test.

Secondary endpoints of response rate measured by Y-BOCS will be assessed in percentages, and significance will be calculated using Fisher’s exact test. Time-to-response analysis will be conducted using the Kaplan–Meier probability.

Overall Sheehan Disability score and Hamilton Depression Rating Scale (HDRS) will be tested. 55-point Likert scales will also be analyzed using a *t*-test, respectively Mann-Whithney-U in case of non-normality distribution. The Sheehan Disability score will be compared for the three individual scores (work, social life, family life) and multiple comparison corrections via the Benjamini Yekutieli procedure. The Mann-Whitney-U test will analyze Clinical Global Impression Scales on a 7-point. Finally, the fMRI study will use the Amplitude of Low-Frequency Fluctuations (ALFF) and regional homogeneity (ReHo) analysis [47] to determine differences in the function of the brain after rTMS.

### 2.12. Missing Data

Missing data will be assessed for each variable. Multiple imputations will be used for primary and secondary endpoints and other potential missing data. In all cases, the model fit will be assessed visually via density plots indicating the distributional discrepancy, i.e., the difference between observed and imputed data.

### 2.13. Ethical Considerations

This study will follow the Ethical Principles for Medical Research Involving Human Subjects and Good Clinical Practice outlined in the Declaration of Helsinki and be considered for ethical approval by the Institutional Review Board Committee at each local site. We intend to register this study in the Clinical trial register under clinicaltrials.gov. Before participating in this trial, each participant will be fully informed about all aspects of the trial, including the experimental procedures, and give written informed consent, and each enrolled patient will receive a copy of the informed consent form.

The investigators are responsible for recording and reporting all serious adverse events (SAEs) that occur throughout the research protocol, starting from the time consent is obtained and continuing through the entire monitoring period. SAEs will be documented using a comprehensive form designed for this purpose, which must be completed, printed, dated, signed, and promptly communicated to the principal investigator. Regardless of when they occur during the protocol, any SAEs suspected to result from the research protocol, where no other reasonable explanation exists, must also be reported.

Adverse events (AEs) that do not meet the criteria for SAEs will be documented only on the patient’s medical card, including details such as the date of onset, characteristics, intensity, duration, possible causes, actions taken, treatments administered, and outcomes. At each evaluation, the investigator will assess whether any AEs or SAEs have occurred and will record all such events on the appropriate case report form (eCRF) page. For each event, the nature, severity, and relationship to the study protocol or treatment will be documented.

### 2.14. Timeline

Participants were recruited through social media, emal invitation, and refrerals from psychiatrist over a three-month period. Overlapping recruitments starts estimated with a total period of 6 months. We will use a two-weeks period for eligibility assessment and included patients will be randomized into two groups, active and control. Baseline assessments, including neuronavigation to locate the dlPFC, motor threshold determination, and functional MRI (fMRI), were conducted over a two-week period prior to intervention. The treatment protocol involved daily sessions, five days per week, for six consecutive weeks. Outcomes were assessed at weeks 2, 4, and 6, with a final post-treatment fMRI conducted after the intervention to evaluate changes in neural activity and clinical parameters (Figure 1).

## 3. Discussion

This study will significantly advance the existing body of research on treatment-resistant OCD by providing robust evidence on the efficacy of low-frequency rTMS targeting the dorsolateral prefrontal cortex (dlPFC). Focusing specifically on patients who have not responded to conventional treatments, this research addresses a critical gap left by previous studies, which often included broader patient populations and were limited by small sample sizes.

Our theoretical study will evaluate the efficacy of inhibition of dlPFC using low-frequency rTMS for 6 weeks compared to a sham intervention in patients with treatment-resistant OCD. Previous studies [33,48] have shown that inhibition of the dlPFC is beneficial for patients with OCD, specifically by reducing OCD symptoms, shown as a reduction in YBOCS score and depression scales. However, these trials had small sample sizes. Additionally, treatment-resistant patients have not always been the focus of previous research. This trial proposes including patients at different levels of non-response, which would theoretically prove the efficacy of TMS in a wide variety of OCD patients. Considering the recent research on this topic, we decided to innovate by choosing a national multicentric study, increasing the sample size considerably, and using real-world experience to improve the generalizability of our results.

This study will contribute to previous research on the topic and will bring feasible and more accessible treatment alternatives for treatment-resistant OCD, increasing the quality of life for patients who struggle with this condition. The strengths of the design include its novelty by being multicentric, avoiding selection and publication bias—detected in previous studies—through randomization and blinding, and having multiple outcome measures [19]. Regardless of the outcomes, this trial proved to explore the efficacy of repetitive transcranial magnetic stimulation (rTMS) as a feasible treatment option for resistant OCD patients.

Innovation is another fundamental aspect of our subject. The research improves the feasibility and accessibility of rTMS as a therapy option by utilizing real-world clinical procedures in a multicentric framework. This novel method guarantees that the study’s results are both technically sound and practically implementable, promoting the incorporation of rTMS into conventional therapeutic practices. Additionally, the nationwide reach of the experiment facilitates the inclusion of varied treatment environments and patient demographics, enhancing the study’s significance and application.

Another new aspect of our study is the use of fMRI to localize the dlPFC. The previous method of the 5 cm rule showed a precision of 13.8% to 54% [22]. This implies that it is highly possible to be in another location, with a different outcome for rTMS in these regions. For instance, Fitzsimmons et al. showed in their meta-analysis that targeting the orbital frontal cortex (next to the dlPFC) with rTMS in OCD patients has no significant effect [15]. Thus, the unprecious localization could explain the wide variety of rTMS use in OCD patients. Using fMRI for localization will therefore improve our knowledge about the specific area of dlPFC as a therapeutic target for rTMS in our patient cohort.

Regardless of the study’s results, this experiment will yield significant insights into the potential of rTMS as an effective therapy option for patients with refractory OCD. Favorable outcomes could validate rTMS as a secure and efficacious alternative to high-dose pharmacological treatments, providing a novel choice for individuals who have depleted traditional therapy avenues. This will not only augment the therapeutic options accessible to physicians but also markedly enhance the quality of life for people enduring treatment-resistant OCD.

In summary, the study’s novel multicentric design, methodological precision, and extensive outcome measurements enable it to significantly influence the domain of OCD therapy research. This research aims to bridge current gaps and surmount the limits of prior investigations, potentially revolutionizing therapeutic procedures and offering significant relief to patients who have not benefited from conventional therapy.

### Limitations

Despite its strengths, the study has some limitations that must be acknowledged. First, it is a theoretical study and has not been implemented into a study. Further the national multicentric design introduces demographic diversity, which enhances generalizability but may also lead to variability that complicates data analysis. Additionally, differences in standard care practices, such as the implementation of cognitive-behavioral therapy across centers, could influence outcomes and serve as potential confounding factors. The intensive six-week treatment regimen may pose logistical challenges for participants, potentially resulting in higher dropout rates and affecting the study’s retention. Furthermore, the absence of a long-term follow-up period limits the ability to assess the sustained efficacy and durability of rTMS treatment effects. These limitations should be considered when interpreting the study results and highlight areas for future research.

## 4. Conclusions

This study represents a pivotal step in advancing both clinical practice and scientific research for treatment-resistant OCD by focusing on the dlPFC as a precise therapeutic target. By exploring the efficacy of low-frequency rTMS as an alternative to high-dose pharmacological treatments, this research has the potential to reconsider the therapeutic landscape for these patients who have not benefited from the conventional therapy approach.

The innovative multicentric design, use of advanced localization techniques by fMRI, and defined methodological framework aim to deliver robust and generalizable evidence that bridges critical gaps left by previous studies. Positive outcomes would establish rTMS as a feasible, accessible, and safer alternative therapy, encouraging its integration into routine clinical care and providing new hope for individuals with treatment-resistant OCD.

Moreover, the findings are expected to contribute significantly to the scientific understanding of rTMS, particularly in refining its therapeutic targeting and improving its efficacy for OCD treatment. Regardless of the trial’s outcomes, this study will generate valuable insights into the mechanisms and applicability of rTMS, enable future research, and further enhance the treatment for OCD. Ultimately, this work seeks to improve patient outcomes and quality of life, underscoring its profound relevance to both clinical practice and scientific inquiry.

## Figures and Tables

**Figure 1 brainsci-15-00106-f001:**
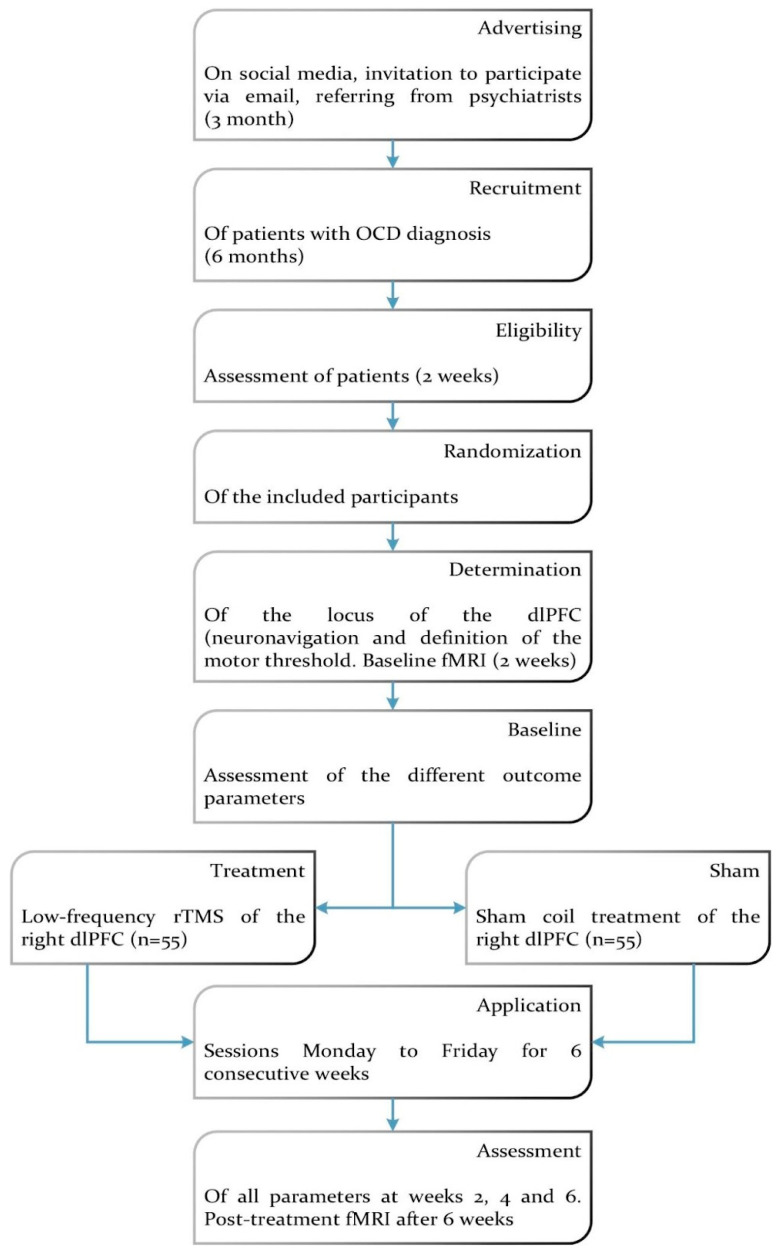
Flowchart of the clinical trial design.

## Data Availability

It is a hypothetical protocol, thus, no new data has been generated.

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
