# Peer review of "MAGNITUDE: Transcranial Magnetic Stimulation for Treatment-Resistant Obsessive–Compulsive Disorder: A Randomized Sham-Controlled Phase II Trial Protocol"

_brainsci, 2025, doi:10.3390/brainsci15020106_

Round 1
Reviewer 1 Report
Comments and Suggestions for Authors
Abstract:
- The objectives are not clearly described.
- The methodology is not clearly defined in the abstract. Who is the target population, and how is the sample selected?
- Although it is a theoretical protocol, it lacks mention of the potential importance of the results in clinical practice.
Introduction:
- The article presents a “Graphical Abstract” before the introduction. This should be included in the methodology section and explained comprehensively.
- The article’s title is “MAGNITUDE: Transcranial Magnetic Stimulation for Treatment-Resistant Obsessive Compulsive Disorder: A Randomized Sham-Controlled Phase II Trial Protocol”, with the stated objective: “seeks to determine the efficacy and safety of low-frequency repetitive transcranial magnetic stimulation (LF rTMS) applied to the right dorsolateral prefrontal cortex (dlPFC) in conjunction with standard therapy for treatment-resistant obsessive-compulsive disorder (OCD) in adults.” The “standard therapy for treatment-resistant obsessive-compulsive disorder (OCD)” should be explicitly included in the introduction.
- The concept of “standard therapy” needs to be better defined, as it is part of the study objectives.
Methodology:
- Although the sample size calculation is mentioned, more details should be included regarding the choice of statistical power (90%) and significance level (α=0.05). Furthermore, the justification for the dropout rate (30%) could be better substantiated.
- While reminders and incentives are mentioned, the methodology does not describe how adherence quality will be assessed beyond tracking attendance.
- More details could be provided on how cases of non-adherence, early dropouts, or incomplete data will be managed.
- It lacks a clear explanation of how participant confidentiality and data management will be ensured. Additionally, there is no reference to an ethics committee submission.
- The section on justification for key methodological decisions, such as the number of sessions and rTMS parameters, should be expanded.
Discussion:
- The discussion could delve deeper into practical implications, such as the feasibility of implementing the protocol in real-world clinical settings.
- It does not explore how this protocol differs from or improves upon previous studies.
Conclusion:
- The conclusion could be more cautious, acknowledging the limitations of a theoretical protocol.
- A more explicit discussion is needed on how the expected results could be translated into clinical practice.
- It does not provide concrete suggestions for future studies, such as evaluating long-term outcomes.
Author Response
We thank you for your time and effort to review our manuscript.
Abstract:
- The objectives are not clearly described.
- The methodology is not clearly defined in the abstract. Who is the target population, and how is the sample selected?
- Although it is a theoretical protocol, it lacks mention of the potential importance of the results in clinical practice.
We appreciate your comments on the abstract to improve it and its informative character. We added our objective to the beginning of another paragraph to enhance its visibility. Further, we added some information on the target population and selection as well as on the potential importance of our protocol, as much as the word limitation for the abstract allows.
Introduction:
- The article presents a “Graphical Abstract” before the introduction. This should be included in the methodology section and explained comprehensively.
We thank you for this note. However, we have to disagree with your suggestion. A graphical abstract is part of the abstract. It should provide a fast overview of the paper and is not a comprehensive figure within the text. As a graphical abstract, we believe it gives a clear, short overview of the protocol.
- The article’s title is “MAGNITUDE: Transcranial Magnetic Stimulation for Treatment-Resistant Obsessive Compulsive Disorder: A Randomized Sham-Controlled Phase II Trial Protocol”, with the stated objective: “seeks to determine the efficacy and safety of low-frequency repetitive transcranial magnetic stimulation (LF rTMS) applied to the right dorsolateral prefrontal cortex (dlPFC) in conjunction with standard therapy for treatment-resistant obsessive-compulsive disorder (OCD) in adults.” The “standard therapy for treatment-resistant obsessive-compulsive disorder (OCD)” should be explicitly included in the introduction.
- The concept of “standard therapy” needs to be better defined, as it is part of the study objectives.
Thank you for this observation. We introduce the topic by explaining how first-line therapies such as CBT and SSRIs are not effective even in most patients with OCD. Furthermore, we added a few sentences on the second and continued lined therapy strategies. We have defined treatment resistance to the standard of therapy as patients who have failed first-line therapies. In addition, we are more specific in the inclusion criteria section. We clarified this important point in the revised version's introduction.
Methodology:
- Although the sample size calculation is mentioned, more details should be included regarding the choice of statistical power (90%) and significance level (α=0.05). Furthermore, the justification for the dropout rate (30%) could be better substantiated.
We appreciate the note to give more justification on the levels for the sample size calculation. However, we do not see the need to justify the significance level of 0.05 as this is the overall convention in medical research. For the Power, we added the half sentence “to reduce type II error,” as this was our intention here. The justification for the 30% dropout rate is due to the combination of dropout rates in rTMS and the OCD patients cohort. We added more substantial justification for these in the paper. In detail: “Due to the low adverse events in rTMS, it has a low dropout rate, from 5.3% in active use to 11.28% in sham use (Berlim 2013). Additionally, the OCD patient cohort has a various dropout rate, from 17.29% in pharmacological trials, over 20.63% in active control like metacognitive therapy, to 23.49% in pill placebo (Johnco 2019). Considering these two, we aim for a 30% dropout rate to ensure we do not lose power during the trial. Thus, our calculation leads to 110 participants overall, with a sample size of 55 participants per group.”
- While reminders and incentives are mentioned, the methodology does not describe how adherence quality will be assessed beyond tracking attendance.
We appreciate your comment. However, we do not assess adherence quality other than attendance. In comparison to drug trials, the participants will receive the intervention in the presence of the applicator. Thus, we do not have to assess adherence quality beyond attendance because we ensure the correct application during the sessions.
- More details could be provided on how cases of non-adherence, early dropouts, or incomplete data will be managed.
We refer to section XIII, where we explained that missing or incomplete data, which are cases of non-adherence and early dropouts, will be assessed by multiple imputations and tested via density plots for the distributional discrepancy.
- It lacks a clear explanation of how participant confidentiality and data management will be ensured. Additionally, there is no reference to an ethics committee submission.
We refer here to section X on data management, where we explained how confidentiality and data management would be ensured by using an independent data monitoring committee without conflicts of interest; confidentiality remained during monitoring, review and deliberations, restricted data access and regularly renewed passwords.
While we already mentioned everything regarding the ethical committee in topic I, we implemented a separate section on ethical considerations in topic XIV.
- The section on justification for key methodological decisions, such as the number of sessions and rTMS parameters, should be expanded.
Thank you for this suggestion. Our justification is that it was used in a clinical trial published by Elbeh et al. 2016, and thus making it comparable. Furthermore, this protocol complies with the safety guidelines for rTMS use, published by Rossi et al. 2012, as we have mentioned.
Discussion:
- The discussion could delve deeper into practical implications, such as the feasibility of implementing the protocol in real-world clinical settings.
Thank you for this comment. Our main focus on this hypothetical study is not the general feasibility of rTMS, as it was already proven by Do et al. (2016) for rTMS and by Fineberg et al. (2023) in a clinical trial in OCD patients. However, we think it can improve feasibility, which we will see when conducting this protocol, especially by the multicentric approach due to “promoting the incorporation of rTMS into conventional therapeutic practices”.
- It does not explore how this protocol differs from or improves upon previous studies.
While we have to disagree that we have not mentioned how it differs from previous protocols (increased sample size, narrow population focusing on treatment-resistant patients only, being multicentric), we have to admit that we have not included the precious localisation technique using fMRI. We added these in the discussion: “Another new aspect of our study is the use of fMRI to localize the dlPFC. The previous method of the 5 cm rule showed a precision of 13.8% to 54%. This implies that it is highly possible to be in another location, with a different outcome for rTMS in these regions. For instance, Fitzsimmons et al. showed in their meta-analysis that targeting the orbital frontal cortex (next to the dlPFC) with rTMS in OCD patients has no significant effect. Thus, the unprecious localization could explain the wide variety of rTMS use in OCD patients. Using fMRI for localization will so improve our knowledge about the specific area of dlPFC as a therapeutical target for rTMS in our patient cohort.”
Conclusion:
- The conclusion could be more cautious, acknowledging the limitations of a theoretical protocol.
We added this concern in the limitation section.
- A more explicit discussion is needed on how the expected results could be translated into clinical practice.
We would like to refer to your questions in the discussion section therefore.
- It does not provide concrete suggestions for future studies, such as evaluating long-term outcomes.
You are correct that we do not provide any suggestions for further studies. However, we are unable to provide this in our theoretical protocol, as we do not have any results on which we can establish further suggestions. Moreover, our protocol itself is a suggestion as a further study.
Reviewer 2 Report
Comments and Suggestions for Authors
The protocol describes a prospective RCT of low frequency repetitive transcranial magnetic stimulation applied to the right dorsolateral prefrontal cortex in conjunction with standard therapy for treatment resistant obsessive-compulsive disorder. The protocol is well described, with novel aspects and theoretically justified.
Minor suggestion:
page 3, line 111: suggest removing the term interesting and replacing with efficacious or something of similiar nature
page 3, line 112: suggest providing further clarification on what is meant by 'no clear evidence', and suggest using the term 'conflicting evidence' as evidence is available, however there is a lack of consensus or consistency in patient outcomes to support an optimal protocol
Page 3, line113: suggest replacing the term 'right point' with 'optimal location' if that is what the authors are referring to
Page 5, line 217- incomplete sentence.
Page 5, line 224, change maniac to manic
Emergency unblinding- suggest revising the protocol for unblinding with improved clarification e.g., in the instance of serious adverse events only, and considering that suicidal ideation may be common in this cohort and may not necessarily warrant unblinding. Also, 'hints of suicidal ideation' does not seem an appropriate metric.
Inclusion criteria- Are patients required to have primary OCD? please specific level of TR- e.g., failure to 2 previous interventions.
Author Response
The protocol describes a prospective RCT of low frequency repetitive transcranial magnetic stimulation applied to the right dorsolateral prefrontal cortex in conjunction with standard therapy for treatment resistant obsessive-compulsive disorder. The protocol is well described, with novel aspects and theoretically justified.
Thank you very much for your positive general assessment of our manuscript.
Minor suggestion:
page 3, line 111: suggest removing the term interesting and replacing with efficacious or something of similiar nature
Thank you for this hint, we adapted it.
page 3, line 112: suggest providing further clarification on what is meant by 'no clear evidence', and suggest using the term 'conflicting evidence' as evidence is available, however there is a lack of consensus or consistency in patient outcomes to support an optimal protocol
Thank you for this notification. We have to admit that “conflicting evidence” is the better term and adapted it.
Page 3, line113: suggest replacing the term 'right point' with 'optimal location' if that is what the authors are referring to
Thank you for this term to enhance our paper.
Page 5, line 217- incomplete sentence.
Thank you for your input. We have rephrased the sentences for clarity.
Page 5, line 224, change maniac to manic
Thank you for pointing out our mistake. We adapted it.
Emergency unblinding- suggest revising the protocol for unblinding with improved clarification e.g., in the instance of serious adverse events only, and considering that suicidal ideation may be common in this cohort and may not necessarily warrant unblinding. Also, 'hints of suicidal ideation' does not seem an appropriate metric.
Thank you for this comment. We are aware that the lifetime prevalence of a suicidal attempt is 1/ 10 (Pellegrini et al., 2020). However, rTMS is so far known as a safe intervention. Treatment-resistant patients have not been assessed as much as other patient groups. Thus, this interaction with neurological brain circuits, in combination with our method to localise the dlPFC correctly, we think that a measurable sign of suicidal ideation makes an emergency unblinding necessary. The SIDAS score gives this measurable sign we will apply in the case, and the limit of 21 indicated already a suicidal preparation or attempt in the past year (van Spijker et al., 2014). However, we acknowledge that the term “hints of suicidal ideation” does not seem appropriate. We changed the wording to “warning signs” based on the National Insitute of Mental Health nomenclature and provided more detailed information on handling these signs. Further changing warning signs of suicidal ideation based on the nomenclature from the National Insitute of Mental Health, like talking about the want to die or feeling unbearable pain (NIMH 2022), will lead us to assess the suicidal ideation attribution with the SIDAS score scale (van Spijker et al., 2014). In case of a score ≥21, this will also lead to unblinding.
Inclusion criteria- Are patients required to have primary OCD? please specific level of TR- e.g., failure to 2 previous interventions.
Thank you for this comment. Since we do not have the requirement for primary oCD only, we haven’t commented on this further in the inclusion and exclusion criteria. The level of treatment failure we defined for our study was listed in the inclusion criteria, last topic, as the current failure of treatment (CBT +/- SSRI) plus the failure of at least one SSRI treatment before. “Limited response (reduction inferior to 25% at YBOCS) to previous treatment defined as patients on maintenance treatment either with cognitive behavioral therapy and/or SSRI and previous failure to respond to at least one SSRI.”
Round 2
Reviewer 1 Report
Comments and Suggestions for Authors
The article is still not ready for publication. I believe it requires further revision, taking into account some of the suggestions made in the previous review.
The Graphical Abstract should be placed in the Methodology section rather than before the Introduction.
In the Timeline subsection, there is a figure that occupies an entire page of the article and is not properly integrated. I recommend presenting this information in text format instead.
Regarding ethical considerations, although adherence to the principles of the Declaration of Helsinki is mentioned, the protocol lacks detailed information on how participant safety and well-being will be monitored. Since this is an interventional study, this aspect should be further developed.
In the Conclusion, the relevance of the study for clinical practice and scientific research is not sufficiently emphasised.
Author Response
The article is still not ready for publication. I believe it requires further revision, taking into account some of the suggestions made in the previous review.
Again, we appreciate your time and energy for the review.
The Graphical Abstract should be placed in the Methodology section rather than before the Introduction.
Since we have a disagreement here, we decided to drop the figure here as we do not see the benefit in including it in the methodology section. In our opinion input, the figure is displayed to be too widespread in the protocol in this section.
In the Timeline subsection, there is a figure that occupies an entire page of the article and is not properly integrated. I recommend presenting this information in text format instead.
We thank you for this comment. We must admit that placing the timeline subsection is not optimal since part of the information is introduced in subsections before and after. Thus, we put it at the end. In our opinion, we do not see any further advancement in this presentation in the text. Much information was provided before; we added this information to the figure regarding the timing. Moreover, assessing this as a figure displays better the information on time flow.
Regarding ethical considerations, although adherence to the principles of the Declaration of Helsinki is mentioned, the protocol lacks detailed information on how participant safety and well-being will be monitored. Since this is an interventional study, this aspect should be further developed.
We appreciate your concerns and added paragraphs for handling adverse and serious adverse events.
In the Conclusion, the relevance of the study for clinical practice and scientific research is not sufficiently emphasised.
We hope the rewritten conclusion enhances the study's relevance.
Round 3
Reviewer 1 Report
Comments and Suggestions for Authors
I consider that, following the revisions made, the article should be accepted for publication.